# Formation of Self-Assembled Anticorrosion Films on Different Metals

**DOI:** 10.3390/ma13225089

**Published:** 2020-11-11

**Authors:** Judit Telegdi

**Affiliations:** 1Institute of Materials and Environmental Chemistry, Research Centre for Natural Sciences, Magyar tudósok körútja 2, 1117 Budapest, Hungary; telegdi.judit@ttk.hu; Tel.: +36-30-4754199; 2PhD School of Material Sciences and Technologies, Obuda University, Doberdó u. 6., 1034 Budapest, Hungary

**Keywords:** self-assembling, amphiphilies, metals, nanolayers, anticorrosion efficiency

## Abstract

The review will first discuss shortly the phenomenon of corrosion and enroll some general possibilities to decrease the rate of this deterioration. The stress will be laid upon the presentation of anticorrosive self-assembled molecular (SAM) layers as well as on the preparation technology that is a simple technique, does not need any special device, and can be applied on different solids (metals or non-metals) alone or in combination. The paper groups the chemicals (mainly amphiphiles) that can develop nanolayers on different pure or oxidized metal surfaces. The history of the self-assembled molecular layer will be discussed from the beginning of the first application up to now. Not only the conditions of the nanofilm preparation as well as their characterization will be discussed, but the methods that can evaluate the corrosion-inhibiting efficiency of the SAM layers under a corrosive environment will be demonstrated as well.

## 1. Introduction

The role of coatings on metals is to protect the solid surface from the influence of the aggressive nvironments. They could be physical effects (like mechanical damage of equipment that diminishes the lifespan, strength, and/or durability) and chemical impacts. To this undesired chemical action belongs the corrosion that leads to degradation of materials, mainly (but not only) of metals by chemical and electrochemical reactions. The results are that materials become run-down from chemical and physical points of view and the degradation costs an enormous amount of money.

### Short Summary of Corrosion

The chemical corrosion is the consequence of the dissolution of metals. In the electrochemical corrosion, the anodic (oxidation) and cathodic (reduction) reactions are coupled, but they take place on different metal surfaces. The summary of the general reaction of a corroding metal is summarized in the next equation, (Equation (1)):M → M^n+^ + ne^−^(1)

Some examples of the corrosion reactions of the mostly used metals are as follows:

In the case of iron: Fe + H_2_O ↔ Fe(OH)_ads_ + H^+^ + e^−^; Fe(OH)_ads_ ↔ [Fe(OH)]^+^ + e^−^. In the anodic reaction, first water molecules adsorb onto the metal surface that alters the product formed on the surface.

When the corrosive deterioration happens according to the so-called catalyzed reaction, the following reactions go on the surface; the second one is the rate-determining reaction: Fe + (FeOH)_ads_ ↔ Fe(FeOH), and: Fe(FeOH) + OH^−^ ↔ [Fe(OH)]^+^ + Fe(OH)_ads_ + 2e^−^. In both types of corrosion mechanisms, the final step is the same—the decomposition of the [FeOH]^+^ species: [Fe(OH)]^+^ + H^+^ ↔ Fe^2+^ + H_2_O. It is important that porous iron oxides and hydroxides are formed on the anode (permeable for aggressive ions) [1,2,3].

When copper is in an aggressive environment, the anodic metal dissolution goes in two steps, according to the following reactions: Cu → Cu^+^ + e^−^; Cu^+^ → Cu^2+^ + e^−^. In the presence of oxygen: Cu + 1/2O_2_ + H_2_O → Cu^2+^ + 2OH^−^, or in chloride ions: Cu + Cl^−^ ↔ CuCl + e^−^; CuCl+ Cl^−^ → CuCl_2_^−^. These reactions influence the metal surface.

When zinc is dipped into acid, the reactions are as follows: on the anode: Zn → Zn^2+^ + 2e^−^; and on the cathode: 2H^+^ + 2e^−^ → H_2_. In the case of hydrochloric acid, the overall reaction is: Zn + 2HCl → ZnCl_2_ + H_2_. In neutral or alkaline solution, the oxygen reduction is the cathodic reaction.

When metal magnesium is dipped into acid, its corrosion follows the reactions: 2Mg → 2Mg^2+^ + 2e^−^; 2Mg^2+^ + 2H_2_O → 2Mg^2+^ + 2OH^−^ + H_2_; 2H_2_O + 2e^−^ → H_2_ + 2OH^−^; 2Mg^2+^ + 4OH^−^ → 2Mg(OH)_2_; MgO + H_2_O → Mg(OH)_2_.

The references give examples on the general corrosion mechanism and on the corrosion processes of different metals [4,5,6,7,8,9,10,11,12,13,14,15,16,17,18]. 

The concrete examples show that the composition and consistency of the corrosion products influence the surface properties. Their loose structure allows the penetration of aggressive ions to the metal surface. For decreasing these undesired processes, anticorrosion methods are applied.

In order to select the most proper anticorrosion technique, it is necessary to become acquainted with the type of the corrosion. The most common ones are the uniform or general corrosion (the whole metal surface is uniformly deteriorated in atmospheric and in acidic environment), the pitting corrosion (rapid penetration of aggressive ions results in small holes and pits), crevice corrosion (due to the retention of water mainly from the atmosphere, the attack takes place at the crevices of the metal surface), stress corrosion (under intense tensile stress, cold or thermal processes, welding), and galvanic corrosion (conductive path connects two metals). All types need special anticorrosion techniques to reduce their undesired negative effect.

## 2. Anticorrosion Possibilities

Various methods have been developed to control the deterioration of metals (e.g., use of dissolved corrosion inhibitors both in aqueous and oily environment or to apply protective coatings in micro- or nanometer thickness). One is the proper selection of materials, and another one is the cathodic protection; the corrosion inhibitors also form a very important group for protection. The role of corrosion inhibiting coatings will be discussed in the next section.

### 2.1. Corrosion Inhibitors

They are chemicals that, added to an aggressive environment at a very small quantity, can react with the metal or with the environment and protect the materials against corrosion by controlling either the anodic or the cathodic processes or both at the same time. They should be effective in a wide range of environmental conditions (high/low temperature, broad pH range and ion concentration, etc.); they also have good solubility and low toxicity, and they should be compatible with the other components in the system [19].

According to the chemicals there are:(a)inorganic inhibitors (chromates, sodium phosphates, polyphosphates, hydrocarbonates, silicates, nitrites) [20];(b)organic inhibitors; they can contain nitrogen (amines, quaternary ammonium salts, diazol/triazoles derivatives) nitrogen and sulfur (thiazole and thiadiazole derivatives), sulfur alone(sulfonic acid derivatives), and oxygen with nitrogen involved into the same molecule (oxazole derivatives) and phosphorous atoms (phosphonic acids where the P atoms are bound to carbon directly: hydroxy phosphonic acids, amino phosphonic acids etc.). In all cases, the lone pair electrons on the hetero atoms as well as π electrons in the molecules are responsible for the inhibitive activity. Organic inhibitors adsorb on the metal surface, forming a film [21,22].

The first and very effective inhibitors were the dissolved chromate salts and the chromate conversion coating, but because of the strong oxidation character and their damaging effects on the human DNA, their use is banned. Other inhibitors that contain mainly N, P, O, S hetero atoms joined to alkyl, aryl, or aralkyl molecular parts could replace the chromate but their effectiveness is not the same as got by chromate. In the last decades, the researchers worked out environmentally friendly inhibitors, applied either as dissolved chemicals (in this case, the inhibitor molecules will adhere/precipitate onto the metal surface, forming a thin barrier that decreases the rate of corrosion) or used in coatings.

### 2.2. Coatings

The effectiveness of the coating methods depends on the structure of the layer formed on the solid surface. If they are compact, the aggressive environment cannot come through the dense layer.

Various techniques have explored to manage/mitigate the corrosion; this is the reason that anti-corrosion coatings have received a lot of attention. There are wide applications of protecting films (conversion coatings, paint with inhibitors, with self-healing and slow release properties, etc.). In order to bring to fruition the replacement of the chromate-free coatings for decreasing the corrosion, several polymers of curing performance were developed since the early 1990s.

Though the history of anticorrosion nanofilms goes back to the beginning of the twentieth century, intensive research started only later on this field. In the last period, the application of thin molecular films—nanocoatings—increased because of their unique characteristics and of the possibility for industrial application, as this technique is simple, and it uses much fewer chemicals.

Thin films that alter the solid surface characteristics are engineered for inhibition of corrosion and biodeposition, and for self-cleaning properties as well as for improved mechanical resistance; all these possibilities make their widespread application clear. 

Types of Anticorrosion Coatings

According to the coating materials, three types of protective anticorrosion coatings are generally mentioned: metallic coatings (produced by electrodepositon, by hot dipping, by vapor phase deposition, etc.); organic coatings (these molecules form a barrier between the metal and the aggressive environment; they could be applied alone or in combination with paints, lacquers, polymers, amphiphilic molecules); inorganic coatings (formed by spraying, by diffusion and chemical conversion). Other possibilities for grouping the anticorrosion coating are smart coatings, hybrid coatings, nanomaterials, and biomaterials. Multi-layer anticorrosion coating systems have primers (a stable surface that locks the upper coat) and topcoats. The technique to produce nanocoatings could be physical and chemical vacuum deposition, plating, spray coating, (thermal/plasma/vacuum plasma/warm/cold spraying) and others like sol–gel coating, dip coating, spin coating, and roll-to-roll coating, to mention only some of them.

One of the mentioned methods that can increase the desired anticorrosion properties is the sol–gel technique, especially when it incorporates inhibitors and/or nanoparticles in a hybrid sol–gel system [23,24]. The sol–gel thin films adhere well to metals as well as to topcoats; this is important from the industrial application point of view.

Other layer deposition techniques are the layered double hydroxide film formation [25] and the layer-by-layer (LBL) technique [26,27,28]. 

## 3. Self-Assembled Nanolayers

The coating techniques summarized previously are effective, but their preparation needs several process steps. In case of the SAM layer formation, not only does the layer formation happen in one step, but the nanocoatings have nanostructure; their thickness is in the range of microns and nanometers, the chemicals used for the formation of the nanolayers are very small. 

For molecular layer deposition two techniques are generally used: the Langmuir–Blodgett method and the self-assembled nanolayer formation. As this paper deals with the self-assembled molecular (SAM) layer technique applicable to decreasing corrosion degradation, henceforth this method will be in the focus.

The SAM surface modification method is generally used to reduce, e.g., corrosion and to mitigate biodeposition, for producing organic thin films on electronic units, for functionalization of a nanostructure on bigger surfaces, etc. [29,30,31,32]. The SAM method was elaborated by different research groups in the middle of the 1980s [33,34,35,36,37,38,39]. In the early experiments, gold and copper surfaces were modified by thiol amphiphiles [40,41]. The research continued to elaborate speciel types of nanolayers, starting from simple amphiphilic molecules and going on to more complicated structures [42].

The formation of SAM takes place when a solid surface is immersed into a surfactant/amphiphile solution (either in water or in organic solvent), that can provide different levels of organization/constructions. The wide range of surface modification can increase the resistance of metals against external impacts. At the same time, the modified surfaces are models for evaluation of adhesion, wetting, friction, etc.

### 3.1. Forces that Ensure the Structure of a SAM Layer

The self-assembling process is spontaneous; the active species in the SAM solution adsorb on the metal surface [43]. Figure 1 demonstrates the formation of the densely packed SAM layer.

The molecules used for SAM preparation could have three building parts. These materials (that are mainly amphiphiles) have a head group (these are ions or polarizable groups that anchor the molecules to the solid surface), a backbone (hydrophobic alkyl, aryl, aralkyl molecular part), and could have a terminal group (they can bind different molecules for building a complex nanofilm).

The head groups can spontaneously adhere by chemi- or physisorption to an oxide-free or to an oxide-/hydroxide-covered metal surface; the hydrophobic molecular parts are kept together via weak non-covalent (hydrogen bonding, electrostatic, van der Waals, ion–dipole, and coordination as well as by π–π stacking) interactions and the consequence is the formation of a well-ordered molecular layer on the solid surface [44,45,46,47]. The self-assembled nanolayers (formed spontaneously from chaotic molecular units and resulted in a close, regular, and well-ordered structure via inter- and intramolecular interactions [48]) could be static self-assembly (no energy dissipation, the layer is in energy minimum) and dynamic self-assembly (when the system dissipates energy via the layer formation).

Summarizing the forces responsible for the SAM film formation and for the SAM structure is as follows: (a) interaction between the head group and the solid surface as well as cooperation between the head groups; (b) interactions among the neighboring hydrophobic molecular parts. All these forces are responsible for the formation of a well-organized molecular layer.

### 3.2. Materials that Influence the SAM Layer Characteristic

The material requirements for the preparation of a well-ordered, densely packed, homogeneous SAM layer are as follows:The head group should have affinity to the metal surface. In certain cases, the presence of oxide layer on metals does not allow the formation of a bond between the head group and the oxide film; these layer-former molecules that adsorb only on oxide-free metal surfaces contain mainly head groups with sulfur atom and the metals that can anchor them are gold, copper, and silver.The length of the carbon chain in the alkyl/aralkyl groups determines the compactness of the SAM layer. Shorter chain (C < 8) does not allow the formation of a densely packed nanolayer.The film will be much more compact if the hydrophobic molecular part (alkyl chain) does not contain any substituent or double/triple bond (especially not near to the head group); in their presence, the chains cannot fit together well, smaller forces will function, and the film will be loose. The consequence is increased permeability.In the case of aryl molecules, the bulky character of a ring has similar impact on the layer quality than the presence of substituent and double/triple bonds.

There are important requirements in deposition of SAM nanolayers. The active molecules anchor to a solid with surface selectivity that depends on the interaction between the surface and the head group of the amphiphiles. The variety of the molecules suitable for SAM formation can modify metal surfaces and qualify these coated metals for application on very different fields.

Other factors that influence the character of SAMs are the sort of metal and its surface roughness (characterized by root-mean-square roughness Rrms) [49]. Generally, a rougher surface allows adsorption of more active molecules, but the irregularity baulks the formation of a densely packed, compact layer. On the other hand, SAMs can be formed on substrates almost of all sizes and shapes.

The hydrophobicity of a coating is influenced by the odd-even effect of the carbon chain (demonstrated by n-alkanethios on gold and silver). The Rrms < 1 can change the effect of the length of the alkyl chain (with its zigzag conformation) and alter the wettability of the SAM coated surface. However, a rougher surface cannot change the wettability of the SAM film [50].

The self-assembled molecular films are mostly one molecule thick, but there are reports on SAM films with several molecular layers that increase their anticorrosion effectiveness. It was demonstrated that stearic acid on copper-nickel alloys deposited in several monolayers enhanced the anticorrosion efficiency and the stability of the film [51].

The molecules covering the solid surface can be perpendicular to the solid or they are tilted. The tilting depends on the number of layers. The formation of SAM phospholipid-like bi- and multilayers of octadecyl phosphonic acid was proved by different techniques when the film was developed from concentrated solution and the solvent evaporation was slow. According to the morphology and structure, this amphiphilic molecule formed a tilted monolayer because of the pressure of the second layer. An interesting observation was that the presence of other molecules with the same active group and shorter carbon chains (C: 8, 14) helped the formation of mixed lipid-type bilayers [52].

Not only the thickness of the layer, but its wettability, which changes with the number of the deposited layers, is important as well. To solve a problem in due measure, sometimes hydrophobic, but in other cases, hydrophilic nanolayer surfaces can take proper effect.

The effectiveness of SAM can be improved by post-treatment of the films via polymerization or restructuring, using heat treatment, irradiation, and UV illumination, to mention only some of the possibilities. Some examples are given in the next references. With increasing UV light illumination time of a SAM layer with alkenyl side chain (i.e., there is a double bound at the end of the carbon chain) and with carboxylic and phosphonic active groups, the compactness of the nanofilm increases because of the formation of a continuous layer. On the other hand, when the head group is thiol, the heating can deteriorate the SAM layer. Gamma irradiation of carboxylic and phosphonic acid SAMs has similar effect like the UV illumination: the SAMs are more resistive to the corrosion as shown in Figure 2.

The original undecenyl phosphonic acid SAM layer’s structure changed via irradiation, and the newer structure can inhibit the pitting effect of the chloride ions even at smaller irradiation. The heating also can increase the anticorrosion efficiency of self-assembled films either by increasing the temperature during the SAM layer formation or at the drying process [53,54,55,56]. However, in some cases, the external impact can destroy the SAM film [55].

Generally, the SAM layers have chemical stability, but they could be sensitive to oxidative influence. Special atmospheric conditions (e.g., ozone) can destroy the thiol SAM nanolayers, when the thiolat head group is transformed to sulfonate. Other researcher explained that the ozonolysis can occur under the influence of UV light at wavelength of 185 nm and 254 nm [57,58,59].

### 3.3. How to Prepare SAM Layers?

The molecular layer deposition could be performed from aqueous and organic solutions. Both possibilities have advantages and disadvantages. Developing molecular SAM layer from aqueous solution has two disadvantages. The layer formation goes parallel with the metal corrosion and the concentration of the active component (that influences the layer deposition rate) is generally low (because of solubility problems). As the concentration has positive effect on the rate of film formation, it is evident that the deposition from aqueous solution needs more time, which increases the possibility of the undesired corrosive deterioration. Generally, organic solvents (e.g. tetrahydro furan, ethanol, chloroform etc.) dissolve much better the layer former molecules producing a more concentrated solution and the deposition rate is enhanced, the corrosion processes eliminated. However, the application of organic solvent—especially at industrial scale—is not environmentally benign.

Another possibility of self-assembled molecular layer formation is when the deposition happens in gas or vapor phase. Though this is a self-limited deposition, in this case the SAM nanolayers have highly uniform nanostructures [60,61]. By this technique, fully uniformly assembled SAM layers are achieved in a much shorter time (in some minutes) than in the case of deposition from liquid. Another advantage of this method is that the thermal stability of vapor-deposited SAMs (as they are mainly organosilane layers) is much better than that of the organic molecular layers [62].

## 4. Techniques Used to Characterize the SAM Layers

The characteristic we want first to learn on a complete SAM layer is the surface coverage that should be perfect, without any irregularities, pores, and failures in order to hinder the penetration of aggressive ions/molecules to the solid surface. Only those nanolayers that achieve these requirements are convenient for industrial application.

The characterization of nanolayers starts during the layer formation and ends when the nanocoating is ready. Other set of experiments will demonstrate the usefulness of the SAM layer in the corrosive environment.

### 4.1. Wettability

In the surface science, the wetting phenomenon is an important subject as the wetting of a surface plays an important role in coating and informs us about the hydrophobic or hydrophilic character of the surface determined by the surface molecules. The principle of the contact angle measurement was already elaborated in the nineteenth century [63,64].

There are two possibilities to measure the contact angle. One is the static version when the shape and the angle of a small, static drop inform us on the surface wettability. The other technique is the dynamic contact angle measurement, when a planar metal coupon, with and without coating is dipped into a solvent (water, organic solvents) [65]. The surface tension of the pure solvent measured by a Wilhelmy plate technique [66] represents the force that pull down the plate. This is equivalent with the contact angle. The evaluation was elaborated by different research groups. Wenzel [67] involved into the contact angle evaluation the role of the surface roughness, and Cassie and his co-workers [68] pointed on the importance of the surface heterogeneity.

The method is important as it can monitor the change in the surface wettability. The contact angle values inform us how the amphiphiles invaded the surface. In the first period of the adhesion, the head groups have a lot of possibility regarding where to anchor, and the molecules are horizontally arranged on the solid surface. With increasing time, more and more head groups adhere to the surface and less and less free places are on the metal surface. At the end of the coating process, the head groups invade the whole surface, and all the hydrophobic molecular parts take up the vertical position. It means that the change in the contact angle allows following the layer formation during the deposition.

### 4.2. Infrared Spectroscopy

The infrared spectroscopy is a useful technique to study nanolayers, as it provides information about the molecular binding, orientation, and conformation. In the case of SAM layer formation, the IRRAS could monitor the time-dependent adhesion of the amphiphilic molecules and inform us about the tilting of the alkyl chains [69]. For the sake of comparison of the usefulness of the infra technique in the characterization of nanolayers, Figure 3 presents the growth of intensity of absorption with increasing layer numbers in the case of Langmuir–Blodgett nanolayers.

Figure 4 demonstrates IRRAS spectra of SAM layers (formed at the same short time) of hydroxamic acids (with alkyl chains: C12–C18) SAM layers.

It is evident that this method can be used perfectly for determination of nanolayer formation time. The curves unequivocally prove the importance of the hydrophobic side chain length (short alkyl chain needs longer time to form densely packed layer.

### 4.3. Sum Frequency Vibrational Spectroscopy

This instrument functions with two laser beams of different wavelengths and informs about the ordering of amphiphiles in the nanolayer, i.e., about the compactness of the nanofilm. It surveys the top of the molecules in the coating, and in the case of all-trans conformation of the carbon chains, it registers the presence of the end groups. When the layer is less densely packed, the sign of the CH_2_ groups appears in the spectrum that indicates the incompactness of the SAM layer [70,71]. Figure 5 displays the SFG spectra of stearoyl phosphonic acid nanolayers (both of the Langmuir–Blodgett and SAM films), and it proves the usefulness of this technique not only for characterization of the nanofilms, but it also helps in determination of the proper layer formation time.

These three methods mentioned above are complementary and characterize a nanolayer very precisely. They inform us about an important parameter: the time and the concentration of the amphiphiles necessary for preparation of a nanolayer with a compact, well-defined structure.

There are other surface characterizing techniques that give important information about SAM layers like the UV-Vis spectroscopy, the photoelectron spectroscopy (XPS), X-ray diffraction, X-ray reflection, scanning electron microscopy (SEM), and scanning tunneling microscopy, etc. An atomic force microscope images the metal surface with and without SAM coatings and allows the visualization of the coating structure; the section analysis of images provides numerical information on the thickness, the roughness, and on the morphology of the film.

## 5. Evaluation of Anticorrosion Activity of SAM Coated Surfaces

To evaluate SAM coated metals under a corrosive environment, the most important techniques are electrochemical methods (electrochemical impedance spectroscopy [72,73,74], polarization resistance method, quartz crystal microbalance, radiotracer methods, nanoscratching, nanoindentation, photoelectron spectroscopy, and Mössbauer spectroscopy [75], but other techniques used every day in corrosion evaluation are also applied (gravimetric/salt chamber method: important information about the pitting, general and intergranular deterioration, inductive plasma coupled mass spectroscopy for detection of metal ions in the corrosive solution) to mention only some of them [76,77]. The corrosion mechanism could be quantified by other techniques [78]. 

All these methods are proper for evaluation of the compactness/usefulness of nanolayers developed on metals from different amphiphiles; they can inform us about the applicability of the layers in a corrosive environment.

## 6. SAM Layers Built from Different Chemicals

The SAM nanolayer is a versatile tool for surface modification and its preparation is simple; this type of surface coverage could be used for very different purposes. The theme of this summary is to show which types of SAM coatings with special head groups can control the corrosive deterioration, and how can the head group and the hydrophobic molecular part influence the anticorrosion efficiency [79]. 

After summarizing the background of the SAM layer formation and evaluation, in the following part the amphiphiles (built-in SAM films) with different head groups will be introduced.

### 6.1. Different Head Groups in the Self-Assembled Molecular Layers

This section discusses the types of nanolayers applied on different metals (on iron, copper, aluminum, and their alloys, which have industrial importance) and their anticorrosion effectiveness in a corrosive environment.

The amphiphiles generally have one head group, but in some cases, bifunctional molecules build the SAM layer. The advantage of these molecules is that one can determine in advance the wettability of the nanolayer and prepare a tailor-made surface. One of the first mentions of application of amphiphiles with two ionic or polarizable groups was in 1983 [80].

#### 6.1.1. Head Groups with Sulfur Atom in the SAM Layers

##### Thiol SAM Layers

The SAM layers formed by alkanethiols have high degree of perfection. The first pioneer experiments already mentioned alkane thiol nanolayers SAM deposited on copper and silver surfaces. The thermal and chemical stability of these 2D systems in aqueous solution or at atmospheric conditions determines their application possibility. The alkane thiols and dialkane thiols are often applied in nanotechnology [81,82].

It is important to emphasize that the thiol group can only anchor to oxide-free metal surfaces. Several papers reported that the alkane thiol molecules (carbon chain length: 8, 12, 16, 18, 22) protect the copper surface dissolution in oxygen-containing environment. Other important observation was that thiols with longer carbon chain built in SAM layers can more effectively inhibit the dissolution of the copper as the compactness of the nanolayer increases with increasing alkyl chain length. A densely packed nanofilm can keep the water molecules, dissolved gases, and ions far from the metal surface. Structural change of the thiol SAMs during corrosion tests was monitored; there was a remarkable shift in the contact angle values measured on thiol nanolayers with shorter carbon chains. In the case of thiol SAM films of a longer alkyl chain, especially in the presence of the other end-group, the contact angle values were not altered significantly. Additional improvement was achieved when other chemicals were parallel applied. An example is when the hydroxyl-docosanthiol and the octadecyl trichorosilan formed a compact layer, where the alkyl chains helped the formation of a double layer [37].

In some cases, a thiol derivative of the trimethoxysilane increased the anticorrosion activity in the presence of chloride ions [83,84,85].

When alkyl silane and thiol molecules form side by side an anticorrosion film, a polymer formed from the silane is bound to the thiol molecular layer that is anchored to the copper surface.

Two cyclohexyl terminated alkanethios and their fluorinated derivatives were also in the focus of a research. By X-ray photoelectron spectroscopy and infrared reflection spectroscopy as well as by wettability measurements the authors ascertained that the structure of the alkyl thiols is much more densely packed than that of the fluorinated derivatives, though the methylene units in the alkyl thiols were less ordered (because of the cyclohexyl group), but the odd-even effect (the influence of the carbon numbers in the chain) was noticeable. These characteristics can influence the compactness of the organic layer and determine the anti-corrosion efficacy [86].

The attachment between the sulfur atom and the copper was illustrated by application of 12-(N-pyrrolyl)-n-dodecanethiol. According to electrochemical and wettability measurements [87], the sulfur atom in this molecule can mainly bond to an oxide-free metal surface. The anticorrosion activity of various alkanthiols in SAMs was demonstrated by XPS study [88], e.g. in the case of octadecanethiol [89,90], and dodecanthiol [91,92]. The influence of the chain length on the SAM layer structure was demonstrated by other authors as well: carbon number: 6, 16, 18: [93], carbon number: 6, 8, 10, 14, 18: [94], carbon number: 10, 12, 16, 18: [95]. They showed that an increasing carbon chain improves the compactness of the nanolayers. Other researchers investigated the influence of an aromatic thiol (thiophenol) on the behavior of an alkane thiol (decanethiol) [96] and a series of the thiophenol derivatives [69]. The conclusions that they drew were that with increase in the alkyl chain length and in the molecular hydrophobicity, the anticorrosion efficiency increases.

The stability of undecane and tri(ethylenglycol) terminated undecane thiol SAM layers in biological media was analyzed in phosphate buffer and calf serum. According to the X-ray photoelectron spectroscopy, the stability of the amphiphiles decreased in phosphate buffer after 21 days; in calf serum, the results were worse because of the desorption of the molecules that was indicated by the S 2p signal [97].

A SAM layer can not only inhibit the corrosive metal dissolution, but this type of film is applicable for metal ion sensing [98]. The p-aminothiophenol SAM layer on gold surface can sense the Co^2+^ ions at very low concentration as was proved by electrochemical measurements. It is well-known that this metal ion is a threat to humans; the cobalt ions analysis can help to trace it in aqueous environment at nM concentration.

##### SAM Layers with Sulphate and Thiosulphate Head Groups

In spite of the common use of sodium dodecylsulphate (which is a well-known surfactant), there are only a few publications on SAMs of amphiphiles with sulphate and thiosulphate head groups. Most of them detail the deposition process and the factors that influence the layer formation, and, in some cases, their application on special fields is mentioned [99,100,101].

It is important to know that the adhesion of these head groups could be influenced by electroactive additives. In these cases, the maximum coverage is obtained when the concentration is below the critical micelle concentration. The amphiphiles will form a layer with a head up-head down orientation [102].

There is a publication on the parallel application of sodium dodecyl-sulfate and ammonium dodecyl-sulfate. Other authors [103] suggest that SAM films of sodium *S*-alkyl thiosulfate (R–S–SO3–Na^+^) with various carbon chain lengths (8, 10, 12, and 14) might be potential alternatives to the traditional thiol-based SAMs, as they form protective layers against copper corrosion. The authors point out that the main advantage of such thiosulfates comparing with thiols is that they are water-soluble and more easily applicable at industrial scale. The results of EIS, IR, contact angle, and XPS measurements indicate that short-chained compounds give lower quality films and provide only modest corrosion protection, while SAM films of compounds with longer chains are more compact and can better protect the metal. This is in accordance with the observation experienced by other SAM films of amphiphilic molecules. The thiosulfate SAMs form less packed, less ordered, and less crystal-like nanolayers than the corresponding thiol SAMS, and they cannot protect the metal surface under a corrosive environment as effectively; these are the reasons why they are less frequently used.

##### SAM Layers of Sulfonic Acids

Unlike thiol (and phosphonic) groups, the sulfonic acid amphiphiles do not form dense, well-structured films on metal surfaces. This observation is supported by results obtained using the Langmuir–Blodgett technique. Most probably, the high solubility of these molecules is responsible for this problem. However, in some cases sulfonic acids combined with a long alkyl chain can form a hydrophobic SAM layer [104].

#### 6.1.2. SAM Nanolayers of Phosphorous Content

##### Phosphate Amphiphiles in SAM Layers

In the introduction, as was already mentioned, hetero atoms are responsible for mitigation of corrosion processes. From this point of view, one of the most important hetero atoms is phosphorous. The phosphoric acid derivatives were used in aqueous solutions to control the metals dissolution. Here are some examples of how its combination with alkyl chains could diminish the undesired metal deterioration.

In this case, the head group is the phosphate, and alkyl chains (with or without substituents) join to it. An example is when dodecyl phosphate and its hydroxy-substituted version covered different metal oxide surfaces (Al_2_O_3_, ZrO_2_, TiO_2_, Ta_2_O_3_, Nb_2_O_5_). A hydrophobic surface was formed in the presence of the dodecyl phosphate (significantly increased water contact angle values proved the presence of the nanofilms). The hydroxyl-dodecyl phosphate could form films on Ta_2_O_3_, Nb_2_O_5_ surfaces, but the surface was changed into a more hydrophilic surface. Co-deposition of these molecules allows the modification of surfaces with tailor-made wettability. Of course, the anticorrosion activity in an aqueous environment needs a hydrophobic surface. The coverage and the orientation of the amphiphiles were determined using different methods [105].

Another example is the use of the naturally occurring phytic acid. The six phosphoric groups that join to a cyclohexyl ring allow a relatively easy metal complex formation on CuNi (70/30) alloy. With increasing phytic acid concentration and dipping time, the rate of the film formation and its quality increased. The anticorrosion activity of this SAM layer in sodium chloride solution proved its usefulness [106].

The titanium and its alloys are often used in medical practice as implants. When this metal is modified by a dodecylphosphoric acid nanolayer, its hydophobicity increases, and the functionalized surface shows an increased lifetime in phosphate buffer [107].

The importance of the alkyl chain length in the SAM layer at this type of molecule was evidenced: the shorter alkyl chain formed a less packed film, while the longer one resulted in a densely packed, compact layer [108].

A very special application of alkyl phosphate layers appeared in some papers [109,110,111]. The authors modified the titanium dental implant surface by phosphoric acid amphiphiles at nanoscale. In this case, not only the corrosion, but the biological deposition was also monitored. The implants have active surface areas that allow the surface modification by the SAM technique.

##### Phosphonic Acids in SAM Layers

In these amphiphilic molecules, the polar/ionic head group is the –PO_3_H_2_ that joins to a carbon atom. This is the difference between the phosphate and phosphonic acid amphiphiles. The phosphates are esters, as they can hydrolyze easily; the phosphonic acids are stable, as the P–C bond is strong.

At the beginning of the research on the SAM layers of this group, the experts revealed that the functional group binds onto a native, more-or-less hydrated metallic oxide layer. This is an acid–base catalyzed condensation which goes in three steps: (1) hydrogen bond forms between the phosphonic acid head group and the metal surface; (2) there is an acid–base catalyzed reaction between the P–OH groups and the –OH groups on the surface, and mono- and bidentate bonds as well as salt and water are formed; 3) the third step produces tridentate bonds via hydrogen bond formation between the phosphoryl oxygen and the surface hydroxyl groups, which is responsible for the self-assembling of the molecule [112,113,114,115]. 

The hydrophobic molecular parts are responsible for the supramolecular formation on the solid surface [116,117]. Figure 6 shows the compact structure of the stearoyl phosphonic acid SAM layers formed on different metals.

The evolution of a well-organized and densely packed phosphonate SAM nanolayer is the consequence of the hydrogen bonding of the acid–base catalyzed interaction as well as of the covalent bonding between the PO(OH)_2_ groups and the metal oxide-hydroxide layers [119,120,121].

Systematic research revealed the importance of the carbon chain length in these molecules as well. As we have seen at the thiol amphiphiles, an increase in the hydrophobic part helps in the preparation of a more stable layer in a shorter time [122].

Another critical point was to see the influence of other hetero atoms in the alkyl chain (e.g., very hydrophobic fluoro atoms instead of hydrogen), and to investigate how the compactness of the layer changes. From the point of view of anticorrosion activity, the increase in the hydrophobicity is important [123].

Not only alkyl-, alkenyl-, and styrene groups containing polymers but fluoroalkyl phosphonic acid SAM layers were also prepared; that proved that not only the increase in the carbon chain length but also the amphiphile concentration and the dipping time significantly enhance the anticorrosion activity of these nanofilms. The corrosion inhibiting efficiency was not significantly influenced by the type of metals [124,125].

Studies on a series of alkyl phosphonic acids (C: 8, 10, 18) and on perfluorodecyl phosphonic acid affirmed the observation of the other researchers [126]. It was interesting that the very effective fluorophosphonic acid inhibits both the general and the pitting corrosion on mild steel, but was less active on aluminum in inhibition of uniform corrosion as the electrochemical and atomic force microscopic corrosion tests proved [29].

On carbon steel, both the fluorophosphonic acid and the undecenyl phosphonic acid could effectively decrease the corrosion rate of the general corrosion, but the undecenyl derivative (that contains one double bond at the end of the carbon chain) was less active in inhibition of pitting corrosion. The post-treatment of the unsaturated bond either with UV light or by irradiation by gamma ray increased the anticorrosion activity because of the formation of a polymer layer over the metals surface [31,70].

When CuNi alloy was covered by the octadecyl phosphonic acid SAM layer, the researchers proved the beneficial effect of heating on the anticorrosion efficacy during the layer formation and in the drying period, as was mentioned earlier [31].

In the case of the mostly investigated octadecyl phophonic acid SAM layer, when it covered titanium, it was demonstrated that the head group is bonded to the metal through oxygen atoms and heat treatment increases the wettability of the nanofilm [43,120].

By application of low voltage scanning electron microscopy, applied parallel with an atomic force microscope, a coated oxidized aluminum was visualized. The experts diagnosed the presence of locally distributed bilayers, but the AFM techniques disclosed the presence of bilayers on the complete surface. They assert that the formation of a uniform SAM monolayer on an oxidized aluminum surface is a complex process [127].

The SAM layers are sensitive to mechanical actions but are much less sensitive to chemical influence, as the nanolayers are well-anchored to the surface. An interesting question was how they behave in flow conditions. It was clearly demonstrated that the phosphonic acid SAM layers developed on CuNi alloy or on stainless steel are stable in natural waters, but their activity decreases in seawater [128].

An interesting application of the phosphonate nanolayer is when this nanofilm helps the bone ongrowth of implants. In this case, a tripeptide (Arg-Gly-Asp) combined with the SAM layer improved the hystomorphometric and mechanical behavior, as was demonstrated in in vivo experiments [129].

A publication mentions the biomechanical investigation of a phosphonic acid amphiphile (11-phosphonoundecanenoic acid) SAM layer formed on a nanostructured TiO surface. After plasma oxidation treatment, they have found bisphosphonate monolayers on the solid surface. The treated surface has higher wettability and shows low surface contact stiffness that can increase the biocompatibility, but it does not allow the formation of a stress shielding effect; this surface modification could improve the utilization of the biomedical implants and facilitate the bone healing [130].

#### 6.1.3. Nitrogen-Containing SAM Layers

The nitrogen with its lone-pair electron can easily bind to metal surfaces. Used as dissolved type inhibitors, they are active in acidic solutions. When they are built in nanolayers, they are active under different conditions.

##### Alkyl Amines in SAM Films

The series of amines with C10–C18 carbon atoms can form a SAM layer on a stainless steel surface when the amino group is anchored only to the pure, oxide-free metal surface. It is interesting that these layers cannot control the corrosion under acidic conditions, as their layer structure is not densely packed [131,132]. The presence of another amphiphile with a different head group (e.g., thiol) can increase the compactness of the layer and consequently, the anticorrosion activity [95]. It is interesting that the octadecanol (though the hydrophobic molecular part is long enough) cannot improve themixed SAM layer characteristic.

##### Amino Acids in SAM Layers

From the point of view of nanolayers the amino acids are important because of the implants that improve humans’ lives. The research on their nanolayers is not very rich, but the publications are important. Among the natural amino acids, arginine (2-amiono-5-guanidinopentanoic acid) is a special one, as it contains four nitrogen atoms. Its SAM layer on an oxide-free copper surface can decrease the dissolution of the metal under acidic conditions [133].

Similar results were mentioned about SAM film formed from histidine (2-amino-3-(1-*H*-imidazole-4-yl) propanoic acid). This molecule, which is of physiological importance, contains one free amino group and a five-member ring with two nitrogen atoms. Its SAM nanolayer can control the corrosion when the environment is acidic, especially in the presence of iodine ions [90,134].

##### Aromatic N-Containing Molecules in SAM Films

It is well known that the aromatic derivatives of tetraazole are good inhibitors in an aqueous environment. There is a report on their application in nanolayers [135]. This publication gives account on the preparation of benzyltetrazole and benzylthiotetrazole SAM layers, and their high anticorrosion activity measured on copper surface is due to the spontaneous adsorption of the tetrazole rings on the metal surface. The presence of the sulfur atom explains the higher corrosion inhibiting activity, which is the result of the plus coupling of the sulfur atom. 

##### SAM Films of Hydroxamic Acids

In these molecules, the N is involved into an acidic group (-CONHOH) from where protons can dissociate. They form a special class of the amphiphiles that can control the corrosion of copper as well as of iron in aqueous solution, even than they have a shorter alkyl chain. When the molecules have a bigger hydrophobic part (longer alkyl chain), they are able to function for molecular deposition and for SAM layer formation. Experiments proved that the layer compactness, its dense structure, increases with increasing deposition time and with an increase in the carbon chain. As the sum frequency vibrational spectroscopy (SFG), the contact angle values, and the atomic force images showed, the best layers with closely packed structures are formed from those amphiphiles that have at least C16 carbon atoms. The film formation is also affected by the temperature; at lower temperature, the film has more compact structure. The adhesion of the head group depends on the mass of the oxide-hydroxide layer as it was displayed by SFG. The natural hydroxides anchor the amphiphilic hydroxamic acid molecules to the metal surface via dismissing the water molecules [70].

Irrespectively from the metal, the alkyl chains in a well-ordered hydroxamic acid SAM layer have *all-trans* conformation, as the XPS and the SFG spectra showed.

The anticorrosion efficiency of hydroxamic acid SAM films was proven by polarization resistance measurements and by electrochemical impedance spectroscopy. These layers can more effectively control the pitting corrosion than the soap-like carboxylic acids [136].

More results on hydroxamic acid nanolayers are summarized in the following references: [104,118,137,138].

As already mentioned in the introduction, the presence of double bonds in the hydrophobic molecular part does not allow the formation of a very well-packed molecular layer. The comparison of the oleoyl and stearoyl hydroxamic acid SAM layer structures and their anticorrosion activities unequally have revealed the role of the double bond: the corrosion inhibiting activity was less than at the other layers, which has almost the same molecular structure except for the double bond [139]. A series of hydroxamic acid nanolayers with different alkyl chains (C: 10, 12, 16, 18) and with a hydroxyl-substituted C18 hydrophobic part was synthesized and characterized. In some cases, when the concentration was high and the layer formation time was long, double layers—that diminished the anticorrosion activity—were deposited [140,141]. An interesting observation was that the hydroxamic acid SAM nanolayers on copper and iron surfaces could not only decrease the corrosion rate, but they also reduced the adhesion of corrosion relevant microorganisms. This could be the consequence of the decreased surface energy (calculated from the contact angle values).

#### 6.1.4. Carboxylic Acid in SAM Nanolayers

The n-alkane carboxylic acids as well as their derivatives (substituted with aromatic rings) can form densely packed nanolayers with well-ordered structure on native metal oxide surfaces. The adhesion of the head groups is similar to that of the other acids. The hydrophobic molecular part is also kept together with weak, non-covalent interactions. The layer deposition depends on the structure of the molecule, on the concentration of the amphiphile, and on the deposition time, similarly to the other film formers.

The self-assembled carboxylic acid molecular layers deposited onto metals can inhibit the corrosion [104,139]. 

Publications appeared about stearic and palmitic acids SAM (formed on mild steel and aluminum) and on their anticorrosion efficiency [142] as well as on the nanolayer of the sodium oleate [143,144] and on a soap-like, substituted alkyl carboxylic acid (12-amino-lauric acid) [145]. All these examples proved the usefulness of these SAM layers as they decreased the metal dissolution, i.e., the corrosion significantly.

In the case of the CuNi alloy, the formation of SAM film from the soap-like stearic acid was in the focus when the researchers investigated the influence of the layer thickness on the anticorrosion efficacy. The experiments proved that a thicker layer (17 nm), which is stable enough, can better control the pitting corrosion that a thin film. The layer thickness could be controlled by the deposition time and the concentration of the stearic acid [51].

Positive impact was demonstrated not only in the case of longer carbon chains but also then, when the molecule has a substituent in ω position [146,147].

On passivated irons, Aramaki and his colleagues analyzed the series of alkane carboxylic acid sodium salts (C: 12, 14, 16, 18) and the 16-hydroxy palmitic acid. They also demonstrated that nanolayer formed of molecules with longer alkyl chain can much better control the corrosion; the presence of the substituent impedes the formation of a regular nanolayer [148].

The n-alkanoic acid substituted with aromatic groups could form an ordered nanolayer on metal surfaces with a native oxide layer. When the silver is not in solid form but as a deposited metal layer, which has a crystal-like structure, the thiolate nanolayer is less stable.

When a special molecule, 4-hexadecyloxybiphenyl-4′-carboxylic acid, forms a SAM layer on silver surface in the presence of H_2_S vapor, a reversible reorganization in the layer structure happens [149].

#### 6.1.5. Silane Derivatives in SAM Layers

In the last several decades, researchers intensively worked on elaboration of the best anticorrosive coatings. Among them, silane-based thin layers turned out to be effective anticorrosive films. This is due to the layer structure. Silanes form SAM layer either in solution or in vapor phase via oxane of Si–O–metal bond formation. The mostly used silanes are the follows: alkyl, fluoroalkyl, cyclic azasilanes, and end-group modified alkyl silanes. The silanes have three hydrolysable groups; they could form siloxane, polymers that are bound to each other and to the solid substrate. The density of the surface metal hydroxyl groups determines the formation of the oxane bond. It is important to select silanes with a proper hydrolysable group in order to get a layer with a well-ordered structure. 

The silane SAM layers have widespread application possibilities (as anticorrosion coatings, in electronic devices, microcontact printing and transistors, for non-volatile memories, etc.).

When the basic material is R (CH_2_)_n_ Si (R’) X2 (R and R’ represent the organo functional groups; X2 is a hydrolysable group; –CH2– is the part of the alkyl chain; Me: metal), the reactions that represent the silane layer formation are as follows: R–Si–OR’ + H_2_O → R–Si–OH + R’–OH; 2R–Si–OH → R–Si–O–Si–R + H_2_O; Me–OH + R–Si–OH → Me–O–Si–R + H_2_O.

The first step is hydrolysis, and then comes the condensation. In these reactions, not only linear but also cyclic siloxanes are formed [150,151]. When the R groups are properly selected, the chemical formation of the Me–O–Si and the surface polymerization enhance the barrier effect. The good anticorrosion properties are due to the formation of the–[Si–O–Si]– closely packed layer.

The anticorrosion effectiveness of a homogeneous silane layer formed from perfluorodecyl trichlorosilane on a metal oxide surface is due to its high hydrophobic character [152].

An example for preparation and application of self-assembled mono- and multilayers of silanes formed on aluminum alloy were described by Wang et al. [153]. The authors used four different alkyl (octyltrimethoxy-, octadecyltrimethoxy-, octadecyltrichloro, and octadecyldimethylchloro) silanes. All of the siloxane layers formed on the aluminum oxide surface increased the anticorrosion efficacy significantly. When multilayers of mixed monolayers are formed, in the course of the self-assembling, the alkylsilane molecules form a mixed layer with opposite molecular orientation. 

Another example is when the siloxan layer is formed from 3-aminopropyltriethoxysilane in combination with grapheme oxide and dopamine (to improve the adhesion). This combination of the components not only decreases the corrosive deterioration but improves the microtribological property [154].

A special superhydrophobic layer is formed on aluminum surface when the trialkoxysilane was applied in combination with stearic acid. This coating protects the aluminum against the atmospheric corrosion [155].

A very special application of a hybrid alkylsilane coating copolymer is that formed from n-decyltriethoxysilane, tetramethoxysilane and 3-aminopropyltriethoxysilane. They produce defect-free layers on magnesium surface. The hydrophobicity of the layer is regulated by the ratio of the components. The specialty of this nanofilm is that not only can it control the corrosion, but it is also bioactive [156].

#### 6.1.6. Mixed SAMs

When SAM layers are composed of two or more components, the deposition happens by either co-adsorption or by sequential adsorption of the amphiphiles. The application of the two components at the same time (that can tune the internal properties) is easy, but it makes the control of the layer formation more difficult. Another problem is when the amphiphilic molecules have different affinity to the solid surface; it could lead to phase separation instead of randomly deposited molecular layer formation. It depends on the process, which is driven either entropically or enthalpically.

In most examples, thiols are applied either with alkanes (of different carbon chains length) or aromatic compounds in combination with aliphatic and alicyclic molecules.

However, there are other possibilities when, e.g., a binary self-assembled monolayer of carboxylic acids, i.e., a Y-shaped aromatic (1,3,5-benzenetribenzoic) acid and a cage-like alicyclic (adamantane) carboxylic acid are applied simultaneously. According to the spectroscopic investigation, the adamantine carboxylic acid molecules are hidden in the densely packed benzene tribenzoic acid monolayer. The interaction of the intermolecular carboxylic group is favored and not the aromatic rings cooperation. The morphology of the film is very similar at wide concentration ratios. It differs significantly from the bimolecular alkane thiol/amides SAM films where the phase separation is mainly due to the H-bonding among the amide groups [157].

Other possibilities for improving the SAMs activity (or vice versa: to enhance the characteristics of the universally applied epoxy resin) is when they are used in mixed form; in these cases, nanolayers help to improve the anticorrosion activity [158].

The special case mentioned in this publication is when the epoxy resin contains zirconium phosphate SAM arranged in nanoplates. This complex coating significantly increases the anticorrosion activity.

Another resolution is when an epoxy layer is combined with tetraaniline that also increases the layer conductivity and the anticorrosion resistance on mild steel [159].

These special layers can control not only the corrosion but the biotribological characteristic on titanium alloys that are applied as implants; the modification could be done by special treatment of the hydroxylated titanium surface, first with aminopropyl triethoxysilane, then with dopamine, and at last, the carboxyl-modified multiwall carbon nanotubes (that have high mechanical properties and proper biocompatibitity) will form a special self-assembled layer on the solid surface. The dopamine forms a transition nanolayer and the three-component coating shows strong adhesion to the titanium alloy; the load-carrying capacity is also improved [32].

## 7. Summary

The intention of this paper was to summarize most of the information on a special nanolayer, i.e., on the self-assembled molecular layers, to demonstrate the conditions for preparation of a well-organized, densely packed molecular nanocoating, and to give examples for their applications, mainly as anticorrosive coatings. Not only were the material requirements brought forward, but the technical problems were also demonstrated to explain how one can prepare SAMs with desired characteristics. 

The materials that are appropriate for nanolayer formation are the so-called amphiphilic molecules with a functional head group and a hydrophobic molecular part. The examples clearly showed, which types of head groups can anchor the amphiphiles and form homogenous layer on oxide-fee or on oxidized metal surfaces. Hot points were the hetero atoms in the head groups (sulfur in thiol, sulphate, thiosulphate, sulfuric acid; nitrogen involved in amino groups, in amino acids, in hydroxamic acids; phosphorous in phosphate, phosphonic acid; and silicon in different silanes, to mention only some of them).For each type of molecules, not only were the adhesion mechanisms, the layer formation, and the forces (that keep the self-assembled molecular layers together) detailed, but examples helped to show their application in a corrosive environment. The SAM organic coating is a very useful technique, as it is simple, does not require special instrument, and the structures of the nanofilms are tailor-made. Before the layer preparation, the composition of the nanolayer could be decided by choosing the proper molecules and the preparation conditions. Post-treatment can improve the compactness and application possibility of the nanolayers.

This technique could be applied not only on small surfaces (e.g., in microelectronics) but on larger ones as well (as in the automotive industry). This is its big advantage, as it does not need an extra facility and equipment, and in that cases when the technology needs an organic solvent, it could be easily recycled.

## Figures and Tables

**Figure 1 materials-13-05089-f001:**
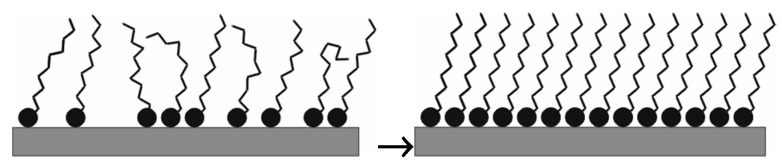
Schema of the time-dependent compact SAM layer formation (with courtesy of L. Románszki).

**Figure 2 materials-13-05089-f002:**
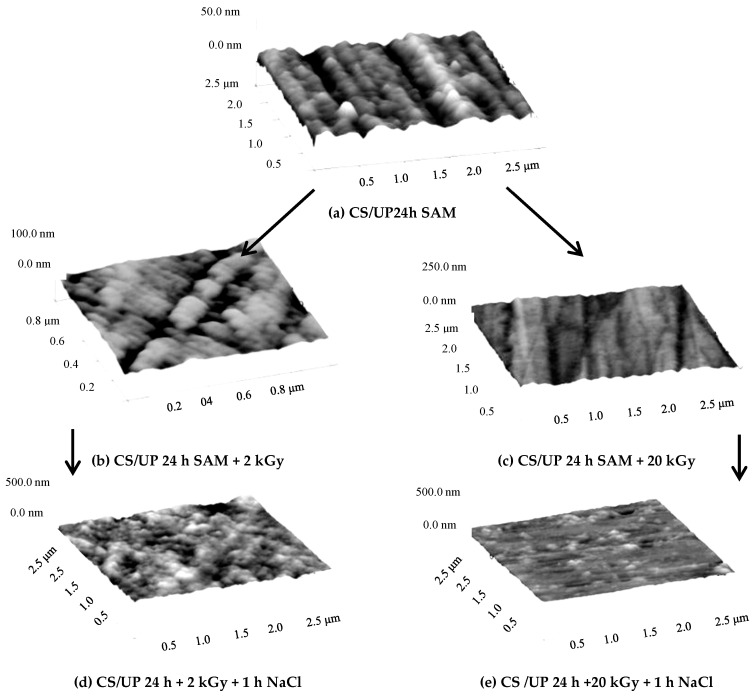
Influence of irradiation on the structure of the carbon steel (CS) surface covered by undecenyl phosphonic acid (UP) SAM layer surface on the anticorrosion activity, visualized by atomic force microscopy (NanoScope III, contact mode, ex-situ experiments), (**a**) carbon steel covered by UP, (**b**) CS/UP irradiated by 2 kGy; (**c**) CS/UP irradiated by 20 kGy; (**d**) sample “b” immersed into NaCl solution; (**e**) sample “c” immersed into NaCl solution; (UP 24 h: undecenyl phosphonic acid SAM layer developed in 24 h, 2 kGy, and 20 kGy: irradiation via ^60^Co gamma source by 2 kGy and 20 kGy) [56] 2018, International Journal of Corrosion and Scale Inhibition (copyright given by Prof. Andreev, the editor of the journal).

**Figure 3 materials-13-05089-f003:**
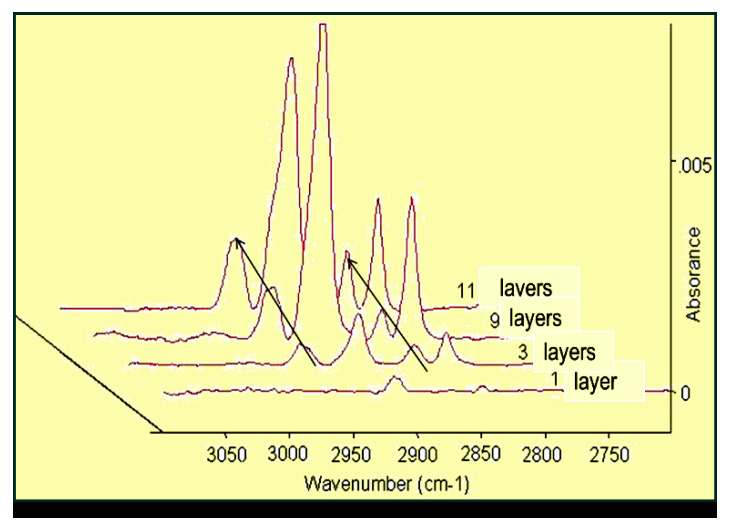
IRRAS spectra of LB nanolayers of stearoyl hydroxamic acid deposited onto copper [69]. Progress in Colloid Polymer Science, Springer, Heidelberg, 2008 (copyright licensed by the Springer Nature).

**Figure 4 materials-13-05089-f004:**
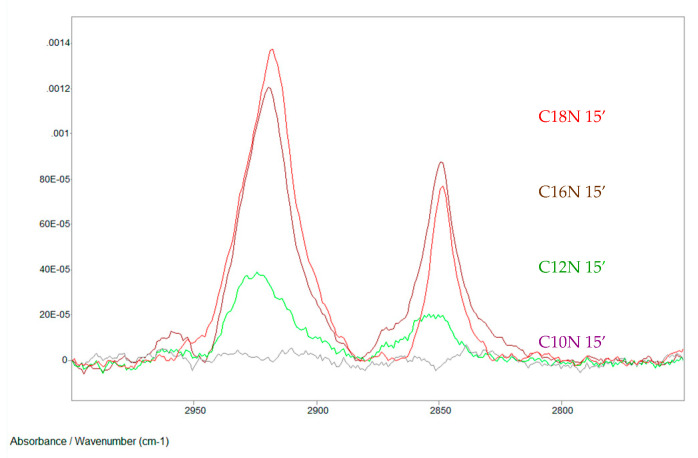
Hydroxamic acids with increasing alkyl chains in self-assembled nanolayers on copper surface; formation time:15 min (C10N: decanoyl hydroxamic acid; C12N: dodecanoyl hydroxamic acid; C16N: palmitoyl hydroxamic acid; C18N: stearoyl hydroxamic acid; the notation e.g. in case of C18N 15’ means that the layer is stearoyl hydroxamic acid SAM formed in 15 min. IRRAS, 0° polarization Nicolet Magna 750 FTIR spectrometer, MCT detector, 512 scans, 2 cm^−1^ resolution, incidence of the IR beam: 68.3°, Au/KRSS polarizer, SPECAC 19650 grazing angle accessory) [69]. Progress in Colloid Polymer Science, Springer, Heidelberg, 2008 (copyright licensed by the Springer Nature).

**Figure 5 materials-13-05089-f005:**
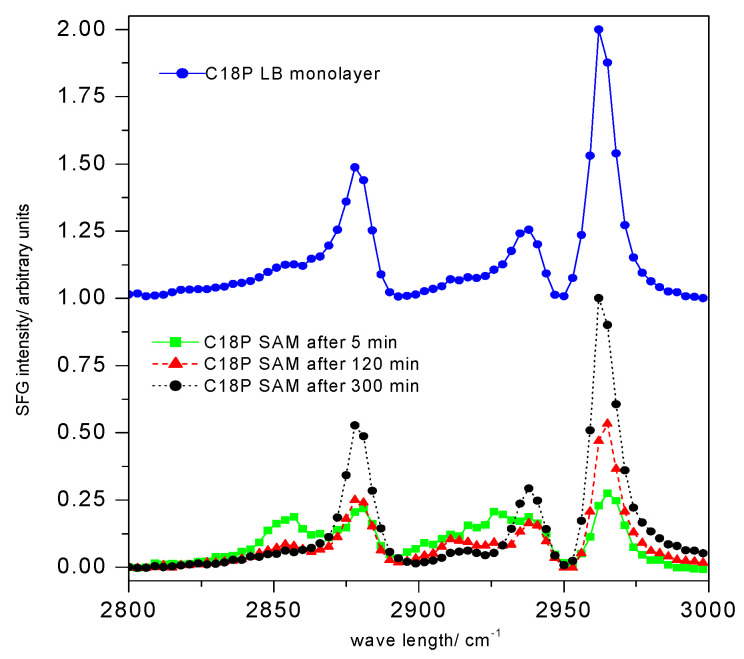
Sum frequency vibrational spectra (SFG) of stearoyl phosphonic acid (C18P) nanolayers (C18P SAM); the upper curve shows the C18P Langmuir–Blodgett (C18P LB) layer’s spectrum; the lower part demonstrates the influence of the stearoyl phosphonic acid SAM layer formation time (5, 120, 300 min) on the spectra; (abbreviation: e.g., C18P SAM 5 means that the stearoyl phosphonic acid SAM layer formed in 5 min) (spectra collected by EKSPLA (Vilnius, Lithuania) sum frequency spectrometer; visible beam: 532 nm generated by output of a Nd:YAG laser (1064 nm), 20 ps pulse width, 20 Hz repetition rate) [69]. Progress in Colloid Polymer Science, Springer, Heidelberg, 2008 (copyright licensed by the Springer Nature).

**Figure 6 materials-13-05089-f006:**
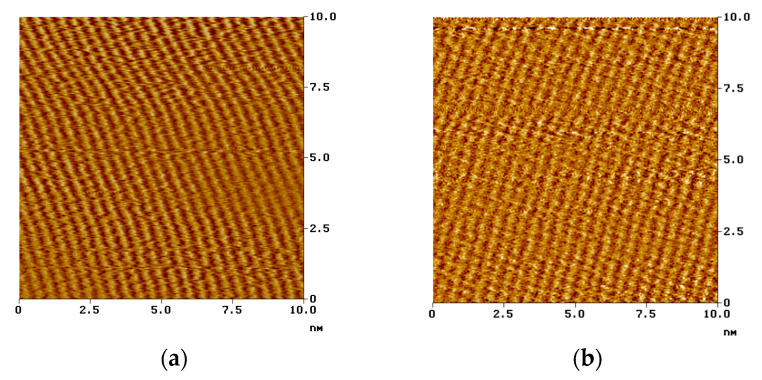
Nanostructure of the stearoyl phosphonic acid SAM layer formed on copper and carbon steel surface, visualized by atomic force microscopy (NanoSocpe III, contact mode, ex situ technique, on air); (**a**): on carbon steel; (**b**): on copper [118]. Corrosion Engineering Science and Technology, 2004 (copyright: Institute of Materials, Minerals and Mining, reprinted by permission of Taylor & Francis Ltd.).

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
