# Peer review of "Formation of Self-Assembled Anticorrosion Films on Different Metals"

_materials, 2020, doi:10.3390/ma13225089_

Round 1

Reviewer 1 Report

The paper reviewed the anticorrosive self-assembled molecular (SAM) layers  on different metals. Generally, the paper were prepared quite good, however some small mistakes present in the present state of the manuscript. There were some structural problems and part of the contents was not very comprehensive.  I suggest the author to make a minor reversion before it was accept for publication. Some qustions were given in following:

  1. Please pain attention to the format of the equations in the part of introcution. For example, the format of equation 1, and the equation in Page 2 line 49. 
  2. Please add some proper referrences in the section of 1. 
  3. In the section of 2, please rearrange the structure of the article of anticorrosion possibilities, Maybe a sub-section of corrosion inhibitors is needed, just as coatings. 
  4.  I suggest the author to state the methods to evaluation of anticorrosion activity of SAM coated surfaces.
  5.  The title is "Self-assembled anticorrosion films deposited onto different metals". Maybe, I think it was better to describe the formation of SAM on different metals respectively, to discussion the differents methods, different groups on various substrate, as the author already given the corrosion introcution of Steel, Cupper, Zinc, Mg, as well as Ti. 

Author Response

Thank you for the constructive remarks and for suggestions for improving the paper.

1. Please pain attention to the format of the equations in the part of introduction. For example, the format of equation 1, and the equation in Page 2 line 49. 

The mistyped parts were corrected:

The equation 1 is corrected:

M ® Mn+  + ne-

The equation in Page 2 line 49 is corrected:

2Mg→ 2Mg2+ + 2e-; 2Mg2+ 2H2O → 2Mg2++ 2OH-+ H2; 2H2O + 2e-→ H2+ 2OH-; 2Mg2++ 4OH-→ 2Mg(OH)2; MgO + H2O ↔ Mg(OH)2

2. Please add some proper references in the section of 1. 

The author added several references on the general corrosion mechanism and on the mechanisms of different metals: [2-18].

3. In the section of 2, please rearrange the structure of the article of anticorrosion possibilities, Maybe a sub-section of corrosion inhibitors is needed, just as coatings. 

The author accepted the reviewer’s proposal and add a part on the corrosion inhibitors.

4. I suggest the author to state the methods to evaluation of anticorrosion activity of SAM coated surfaces.

This topic is summarized in the section 5. Now I added some literature to complete it.

5. The title is "Self-assembled anticorrosion films deposited onto different metals". Maybe, I think it was better to describe the formation of SAM on different metals respectively, to discussion the differents methods, different groups on various substrate, as the author already given the corrosion introcution of Steel, Cupper, Zinc, Mg, as well as Ti. 

The author changed the title: “Formation of self-assembled anticorrosion films on different metals”

Reviewer 2 Report

  1. Please check the reaction on Mg p.2 lines 49.
  2. Any reference for simplified corrosion reactions is needed in Introduction.
  3. Any references for coating deposition methods are also desired to provide as the authors mention sol-gel and LBL techniques (p.3 lines 89-98).
  4. Figure 2 and some others. Please specify the techniques which were used to create images in the figure captions.
  5. If possible, please add some description of the scalability of SAM techniques for industrial applications. 

Author Response

The author say thanks for the proposed corrections that can improve the paper.

1. Please check the reaction on Mg p.2 lines 49:

The equation is corrected.

2. Any reference for simplified corrosion reactions is needed in Introduction.

The author added references to present the corrosion mechanism generally, on iron and on other metals: references [2-18].

3. Any references for coating deposition methods are also desired to provide as the authors mention sol-gel and LBL techniques (p.3 lines 89-98).

The author completed this part with proper references.

4. Figure 2 and some others. Please specify the techniques which were used to create images in the figure captions.

The author completed the captions with the missing details.

5. If possible, please add some description of the scalability of SAM techniques for industrial applications. 

The author add some sentences at the end of the summary that show how useful this technique is.

Reviewer 3 Report

In this contribution the author provides a nice collection and informative data on Self-assembled anticorrosion films. However the introduction section of the review is not complete and should be elaborated more in order to provide the reader the idea behind this review. Also some recent articles in this field should be mentioned in the introduction to show the importance of the studied subject. The author is encouraged to use the following articles beside many more recent articles in this field:

  • "Structure, mechanical properties and corrosion resistance of amorphous Ti-Cr-O coatings", Surface and Coatings Technology, 2019, 374, 690-699.
  • "Mechanical properties and microstructural stability of CuTa/Cu composite coatings", Surface and Coatings Technology, 2019, 364, 22-31

In my point of view this review can be published after this minor revision on the introduction section.

Author Response

In this contribution the author provides a nice collection and informative data on Self-assembled anticorrosion films. However the introduction section of the review is not complete and should be elaborated more in order to provide the reader the idea behind this review. Also some recent articles in this field should be mentioned in the introduction to show the importance of the studied subject. The author is encouraged to use the following articles beside many more recent articles in this field:

The author say thanks for the proposed corrections and gave several proper references to the paper, among them recent papers.

Reviewer 4 Report

The review describes nanocoatings, prepared by a simple and relatively cheap self-assembling method, that can be successfully applied as anti-corrosion protection for many metals and alloys of technologic importance. Although it is mainly focused on their anticorrosion function, other applications such as bioactivity of implant materials were mentioned.

It is clearly written and well structured. According to the reference list, the review is based on recent scientific data.

Therefore, I can recommend publishing in the present form.

Author Response

The author expresses her thanks for the very positive assessment of the paper.